# A Novel Approach to Match Individual Trees between Aerial Photographs and Airborne LiDAR Data

Yi Xu [1,*], Tiejun Wang [1], Andrew K. Skidmore [1,2] and Tawanda W. Gara [3]

1 Faculty of Geo-Information Science and Earth Observation (ITC), University of Twente, P.O. Box 217, 7500 AE Enschede, The Netherlands; t.wang@utwente.nl (T.W.); a.k.skidmore@utwente.nl (A.K.S.)
2 School of Natural Sciences, Macquarie University, 12 Wally's Walk, Sydney, NSW 2109, Australia
3 Department of Environmental Science and Management, California State Polytechnic University Humboldt, Arcata, CA 95521, USA; tawanda.gara@humboldt.edu
* Correspondence: y.xu-3@utwente.nl

**Abstract:** Integrating multimodal remote sensing data can optimize the mapping accuracy of individual trees. Yet, one issue that is not trivial but generally overlooked in previous studies is the spatial mismatch of individual trees between remote sensing datasets, especially in different imaging modalities. These offset errors between the same tree on different data that have been geometrically corrected can lead to substantial inaccuracies in applications. In this study, we propose a novel approach to match individual trees between aerial photographs and airborne LiDAR data. To achieve this, we first leveraged the maximum overlap of the tree crowns in a local area to determine the correct and the optimal offset vector, and then used the offset vector to rectify the mismatch on individual tree positions. Finally, we compared our proposed approach with a commonly used automatic image registration method. We used pairing rate (the percentage of correctly paired trees) and matching accuracy (the degree of overlap between the correctly paired trees) to measure the effectiveness of results. We evaluated the performance of our approach across six typical landscapes, including broadleaved forest, coniferous forest, mixed forest, roadside trees, garden trees, and parkland trees. Compared to the conventional method, the average pairing rate of individual trees for all six landscapes increased from 91.13% to 100.00% ($p = 0.045$, *t*-test), and the average matching accuracy increased from $0.692 \pm 0.175$ (standard deviation) to $0.861 \pm 0.152$ ($p = 0.017$, *t*-test). Our study demonstrates that the proposed tree-oriented matching approach significantly improves the registration accuracy of individual trees between aerial photographs and airborne LiDAR data.

**Keywords:** remote sensing; multimodal images; object-based image analysis; spatial mismatch; registration noise; intersection over union





## 1. Introduction

Information on individual trees is essential for forest management, biodiversity conservation, and assessment of ecosystem services [1–3]. Meanwhile, as a meaningful unit of analysis at a fine scale [4], individual tree level information is crucial for accurate estimation of essential biodiversity variables (EBVs) [5–7], such as biomass [8–10], structural and chemical properties [11,12], as well as functional diversity [13,14]. However, traditional tree inventory tends to be carried out at the plot or stand level [15–18] since obtaining individual tree level information over large areas is usually impractical or impossible via ground survey methods due to extremely high cost and labor-intensive fieldwork [19–21].

To complement in situ measurements, remote sensing has been commonly used to study individual trees at a local or landscape level over the past two decades [22–24]. A critical step before studying individual trees with remote sensing data is delineating individual tree crowns [22,25,26]. High spatial resolution optical remote sensing data (e.g., satellite images, aerial photographs, and drone images) have a proven potential to effectively segment individual trees [3,23,27–39]. However, segmenting individual trees

using optical images often leads to over- or under-segmentation of trees as objects in different scenarios, caused by spectral heterogeneity, noise pixels (intensity variation), shadows, as well as observation and illumination angles for imaging [30,31,33,40].

Light detection and ranging (LiDAR) can render three-dimensional (3D) structural information of trees [41–44]. Recently, with the rapid advancement of LiDAR sensors and the availability of LiDAR data, LiDAR technology has gradually become an indispensable active remote sensing technique for delineating individual trees [45–48]. Small-footprint airborne [49–52] and drone (UAV)-based LiDAR data [53–55] are the primary sources of LiDAR data that have been widely used to segment individual trees. Compared to optical sensors, LiDAR is not impacted by illumination artifacts such as the shading of shorter trees by their taller neighbors [25,56,57]. For that reason, the problem of under-segmentation has been effectively resolved with LiDAR by some existing individual tree segmentation algorithms [58–60]. However, these algorithms do detect multiple peaks in tree crowns, resulting in varying degrees of over-segmentation [61,62]. To address this problem, recent studies have attempted to improve the accuracy of individual tree delineation by integrating optical and LiDAR data [63,64]. These studies have shown that the accuracy of delineating individual trees can be considerably improved by fusing both spectral (optical images) and structural features (LiDAR data). Consequently, the fusion of multimodal remote sensing data for delineating individual trees has gained increasing attraction in recent years.

Local mismatches (misalignments) between images are often overlooked when integrating multimodal remote sensing data (e.g., fusion of optical and LiDAR data) to identify individual trees [65,66]. However, local mismatch still occurs and is referred to as registration noise/error [67,68] due to dissimilarities in acquisition circumstances and imaging principles of the sensor with different modes [69,70]. Moreover, images in different modes are more likely to cause complicated nonlinear deformation and displacement of ground objects between images [71–73]. The positional discrepancy between multimodal datasets has a significant effect on subsequent applications through error propagation [74–77]. Area- and feature-based matching are two methods commonly utilized to align remotely sensed images [78].

The area-based matching processes the intensity of corresponding image areas and searches for the best matching similarity in the window template, but the classic methods are not robust enough for multimodal image with apparent radiometric difference [79,80]. Although some normalized cross-correlation (NCC) (one of area-based methods) variants like automatic registration of remote sensing images (ARRSI) and orientation phase consistency histograms (HOPC) were proposed to allow NCC framework to realize image registration with nonlinear radiometric difference [81,82], they were limited to invariance of intensity mapping matching process and specific geometric and radiometric constraints, respectively [81,83]. In terms of feature-based matching, the most common method is to explore tie points between the reference image and candidate image by manual selection or automatic techniques [84]. Yet, the manual selection is labor intensive, time consuming, and prone to subjective errors, which is only suitable for small areas or medium- and low-resolution remote sensing imagery with spatially consistent offset. In terms of automatic approach, the scale-invariant feature transform (SIFT) [85] and variations thereof [86–89] are the most commonly used automatic algorithms [90,91]. These algorithms aim to detect the homonymy points based on local features in images, which is hence difficult or even not available to be applied in multimodal images due to the influence of nonlinear radiometric differences [73,92].

In recent years, researchers have attempted to use deep learning techniques to register multimodal images [93]. Two main strategies have been explored to leverage deep learning for image registration [72]. The first strategy is deep iterative methods that employ neural networks to compute advanced similarity metrics between image pairs and guide iterative optimization for accurate registration [94]. Examples of deep iterative methods include stacked autoencoders [95–97], convolutional neural networks (CNNs) [98–100], and reinforcement learning [101]. The second method is deep transformation estimation that

directly predicts geometric transformation parameters or deformative fields. This method can be supervised or unsupervised, depending on the training strategy [102–105]. Supervised methods need ground-truth data and use various architectures and loss functions for training. Unsupervised methods rely on traditional similarity metrics and regularization for loss functions. They also use spatial transformer networks for end-to-end transformation learning and adapt to multimodal images. Deep learning-based methods have achieved remarkable progress in remote sensing image registration, but they also face challenges to practicality and generalizability [106]. They require substantial labeled data and computational resources for training and inference. However, traditional approaches, which are known for their efficiency and simplicity, struggle to match the high accuracy showcased by deep learning models. Therefore, considering the trade-offs between the advantages and challenges of deep learning, it is critical to further refine and expedite image registration processes, especially for multimodal images.

In this study, we propose a novel tree-oriented approach to match individual trees between aerial photographs and airborne LiDAR data. To achieve this, we first leverage the maximum overlap of the tree crowns in a local area to determine the correct and the optimal offset vector, and then use the offset vector to rectify the mismatch on individual tree positions. We used pairing rate (the percentage of correctly paired trees) and matching accuracy (the degree of overlap between the correctly paired trees) to assess the effectiveness between our proposed approach in matching individual trees and compare it with a commonly used automatic image registration method.

## 2. Methods

### 2.1. Proposed Tree-Oriented Matching Approach

When processing images, pixels are first grouped as an object, based on either spectral similarity or external variables (such as ownership and geological unit) as a new analysis unit, which is a more effective and meaningful alternative way to studying targets than using pixels directly [107–109], i.e., object-based image analysis (OBIA) [107,110]. The results of delineating an individual trees boundary fits the concept of OBIA, and some structural traits may be more easily retrieved at an individual tree level (e.g., tree height, crown size, and the diameter at breast height) [111,112]. Therefore, we proposed an individual tree-oriented matching approach to improve the matching accuracy of individual trees from aerial photographs and airborne LiDAR-derived CHMs. The flow diagram of the approach is shown in Figure 1 and followed by a detailed description of each step (Algorithm 1).

### 2.2. Automatic Image Registration Workflow in ENVI

To compare with the traditional method, we also used the *Automatic Image Registration Workflow* module in mainstream commercial software ENVI (version 5.6.3 (c) 2022 L3Harris Geospatial Solutions, Inc., Manila, Philippines) to geometrically align two images [113]. We chose the mutual information matching method (developed for images with different modalities) to automatically generate tie points and kept other parameters as default. After tie points were generated, we removed those with clear errors and then registered the warp image with the remaining points. Finally, we delineated bounding boxes of individual trees from the registered images to compare the effectiveness of both methods.

---

**Algorithm 1:** The pseudocode of proposed tree-oriented matching approach.

---

Matching individual trees is
**Initialize**:
minimum and maximum crown area ratio: *min_car*, *max_car*; maximum offset: *max_offset*
**Input**: canopy height models (CHMs) # reference dataset;
        aerial photographs (ARPs) # candidate dataset
**# Step 1**: Mark individual trees with bounding boxes
(The bounding boxes were delineated manually in this study)
*CHM_trees <- mark_trees(CHMs)*
*ARP_trees <- mark_trees(ARPs)*
**# Step 2** Iterate over a tree in CHMs as a reference tree (Figure 1a)
*for reference_tree in CHM_trees:*
     *search_center <- reference_tree.center*
     **# Step 3**: Choose candidate trees according to *search_center* and *max_offset* (Figure 1b)
     (To filter those trees whose center exceeds the maximum offset of the reference tree)
     *candidate_trees <- ARP_trees.within_radius(search_center, max_offset)*
     *reference_trees <- CHM_trees.within_radius(search_center, max_offset)*
     **# Step 4**: Filter candidate trees based on crown area ratio (Figure 1c)
     (To filter those trees whose crown area is much larger or smaller than that of the reference
tree)
     *candidate_trees <- candidate_trees.filter(min_car, max_car, reference_tree.crown_area)*
     **# Step 5**: Calculate offset vectors for each candidate tree and rectify tree locations (Figure 1d)
     (To rectify the candidate trees according to the offset vectors)
     *for candidate_tree in candidate_trees:*
       *offset_vector <- reference_tree.center() - candidate_tree.center()*
       *candidate_rectified_trees.append(candidate_trees.rectify_location(offset_vector))*
     **# Step 6**: Calculate NIoU and select the correct offset vector (Figure 1e)
     (To choose the correct offset vector for the currently selected reference tree that maximizes
the sum of NIoU between the selected reference trees and the candidate trees)
     *correct_offset_vectors.append(max(sum(NIoU(reference_trees, candidate_rectified_trees))))*
**# Step 7**: Rectify tree locations based on the correct offset vectors (Figure 1f)
(To rectify the trees in aerial photographs with the correct offset vectors)
*for correct_offset_vector in correct_offset_vectors:*
     *ARP_rectified_trees.append(ARP_trees.rectify_location(correct_offset_vector))*
**# Step 8**: Determine the final offset vector (Figure 1g)
(To determine the final offset vector that maximizes the sum of NIoU between individual trees in
CHMs and the rectified trees in aerial photographs)
*final_offset_vector <- max(sum(NIoU(CHM_trees, ARP_rectified_trees)))*

---

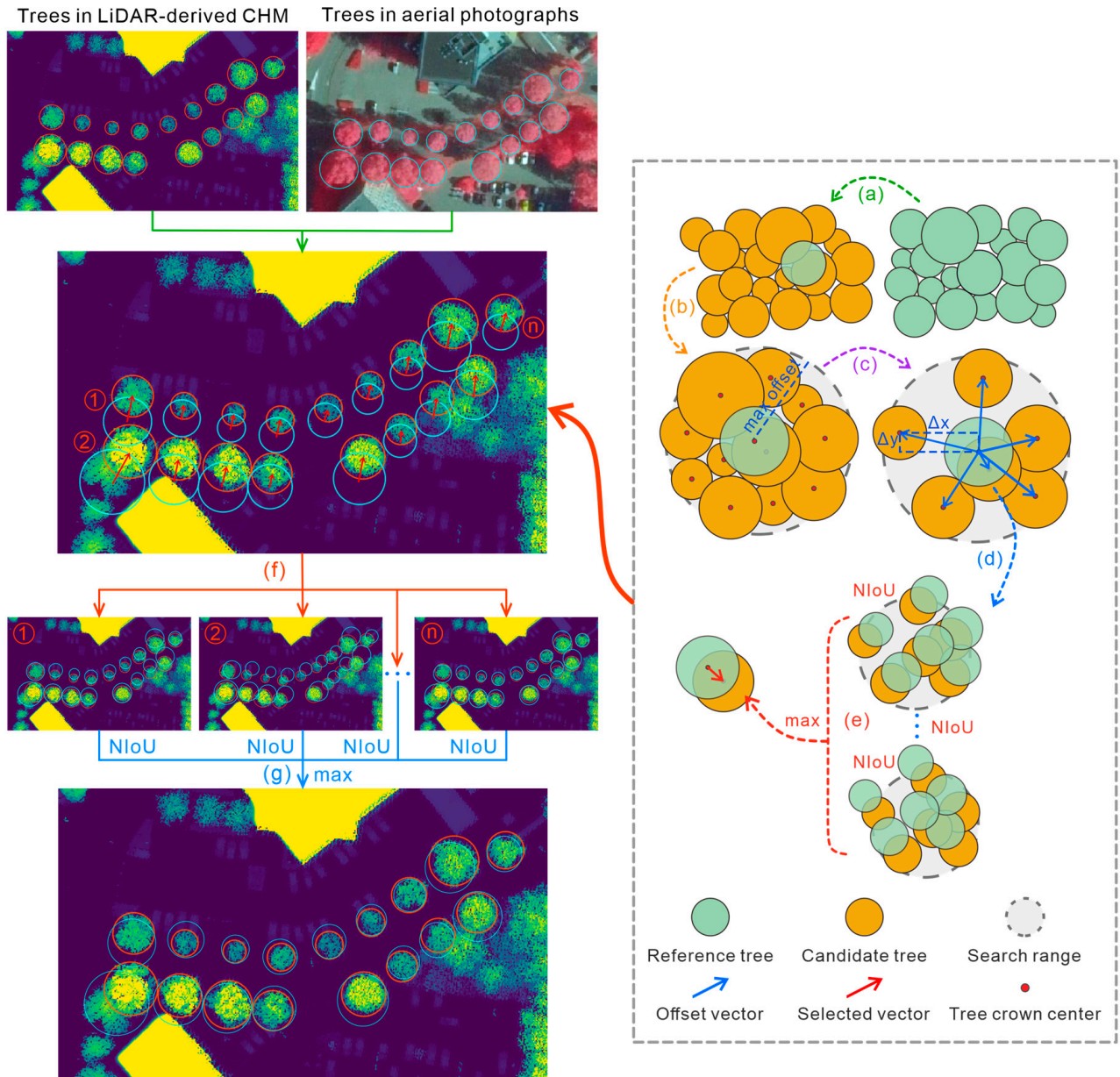

**Figure 1.** The flow diagram of the proposed tree-oriented matching approach.

## 3. Experiments

### 3.1. Experimental Site

The experimental site is Arnhem, the Netherlands (Figure 2). Arnhem is a city and municipality situated in the central part of the Netherlands, containing many parks and forests. The north corner of the municipality is part of the Veluwe—the largest forest area in the Netherlands. Following the definition of the United Nations' Food and Agriculture Organization (FAO) [114] and CORINE land cover classification scheme [115], we divided the forest landscapes in the study area into broadleaved forest (>75% cover of broadleaved trees), coniferous forest (>75% coniferous trees formation), and mixed forest (composed of more than 25% coniferous and broadleaved trees). Furthermore, due to some differences in the layout, density, species richness, and context of groups of individual trees in the city, we also chose three more landscape types—roadside trees, garden trees, and parkland trees. Specifically, roadside trees, regularly lined the road, are often composed of the same tree species with similar height and crown size in a local area (4–16 trees/100 m of linear

feature [116]); garden trees, planted in the garden in front or behind a house or buildings, are often of various species and vary in height and crown size (but contain > 25 trees ha$^{-1}$ [117]); and parkland trees, planted based on the layout of parkland, are species abundant, usually having large widely spaced trees separated by grass areas (>18 trees ha$^{-1}$). Samples of individual trees were chosen from these six landscapes to represent the diversity and complexity of tree distribution, with the level of tree cover used to assess the effectiveness and robustness of our proposed approach.

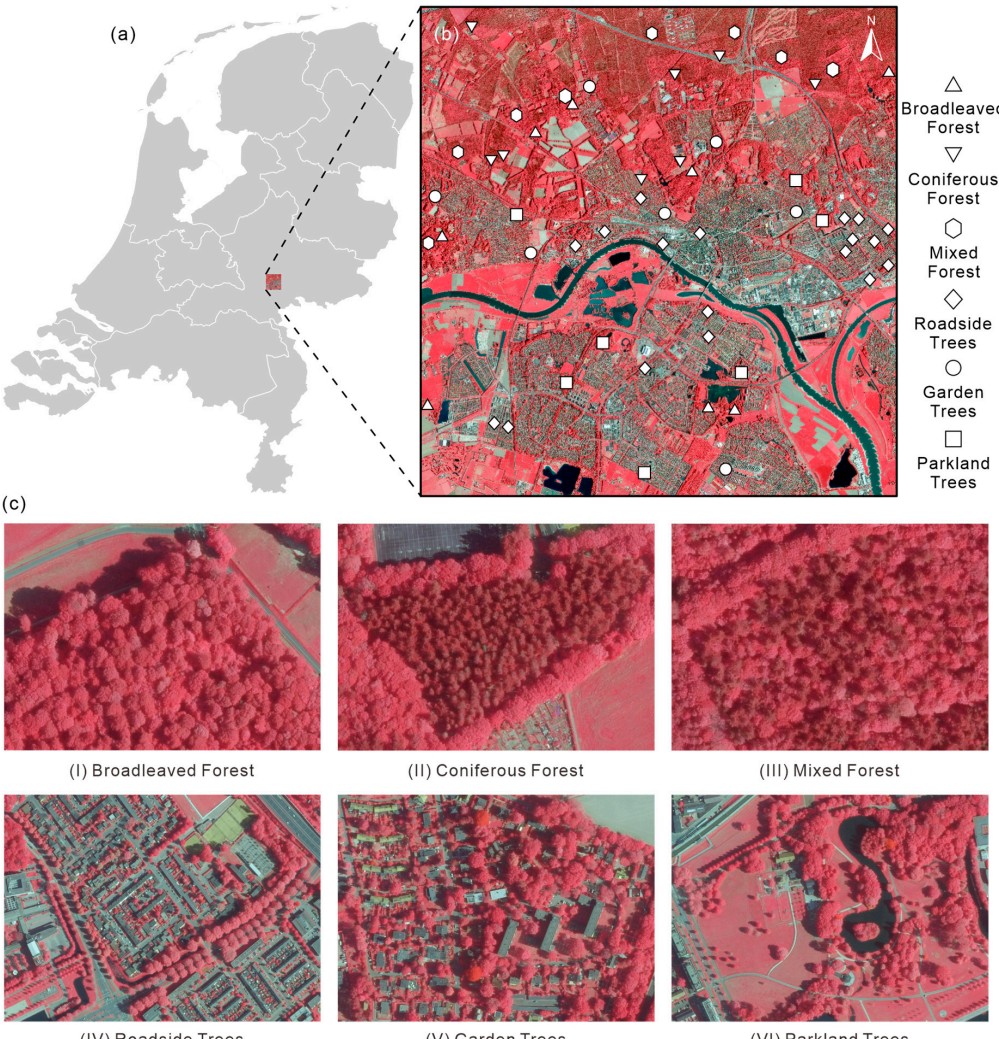

**Figure 2.** Location of the experimental site and the distribution of sample trees within the six typical landscape areas. (**a**) The location of the city of Arnhem in the Netherlands, (**b**) the distribution of sample trees selected in the typical landscape areas, and (**c**) sample landscapes from the aerial photographs of the study area.

### 3.2. Aerial Photographs

The Dutch government conducts country-wide aerial photography campaigns twice a year with one winter (leaf-off season) and one summer (leaf-on season) since 2012. The aerial photographs acquired in summer with four bands (red, green, blue, and infrared) have a spatial resolution of 25 cm, while the aerial photographs of winter with a spatial resolution of 8 cm have only three bands (red, green, and blue). Both datasets are openly accessible through the web map service (WMS) (opendata.beeldmateriaal.nl). The aerial photographs used in this study were captured during the summer in 2020. There are two reasons for choosing the aerial photographs acquired in 2020. On the one hand, higher image clarity and stability help us delineate tree canopies (the comparison is shown in

Figure S1). On the other hand, it can better simulate real-world applications since data from different sensors are often acquired on different dates and mounted on various platforms.

### 3.3. Airborne LiDAR Data

The airborne LiDAR point cloud data were from the 3rd national airborne LiDAR flight campaign in the Netherlands (i.e., Actueel Hoogtebestand Nederland 3, AHN3), which can be openly accessible via the online repository PDOK (app.pdok.nl/ahn3-downloadpage). The AHN3 data of our study area were acquired during the leaf-on season in 2018 (ahn.nl/kwaliteitsbeschrijving). The average point density of AHN3 is between 6 and 10 points m$^{-2}$, and the area of individual trees can reach approximately 40 points m$^{-2}$ owing to the penetration capability of LiDAR.

### 3.4. Generation of Canopy Height Models from Airborne LiDAR Data

Canopy height models (CHMs) derived from LiDAR point cloud data have been commonly used to generate the boundaries of individual tree crowns [49,118,119], which is obtained through cutting a digital surface model (DSM) with a digital terrain model (DTM) [120–122]. To maintain the same spatial resolution as aerial photographs, we processed AHN3 LiDAR point cloud data to generate CHMs with a spatial resolution of 25 cm. To generate CHMs that could show clearer boundary of tree crowns, we conducted the following processing chain for the LiDAR point cloud. The first step was to normalize the LiDAR data, aiming to eliminate the influence of topographic relief on the elevation value of point, and those points classified as "unclassified" were extracted from the original AHN3 data. To perform normalization, the elevation of the nearest ground point to each point is subtracted from its elevation value. Then, the points with elevation less than 25% of the mean elevation in clipped point cloud dataset were eliminated, and the outliers were removed [123,124]. Finally, inverse distance weighting (IDW) function was chosen to generate the CHMs, a widely employed deterministic model in spatial interpolation that relies on the first law of geography [125–127]. The above processing was all implemented with the software LiDAR360 (version 4.1).

### 3.5. Delineating Individual Tree Crowns from Aerial Photographs and CHMs

In this study, we used manually delineated individual trees as samples to verify and test our proposed approach, which can effectively avoid the omission and commission error of automatic detection algorithms for tree detection. To improve the reliability of the crown boundaries in aerial photographs and CHMs, airborne images with higher spatial resolution (8 cm) and LiDAR point cloud data with 3D spatial structure information were employed as reference images. To guarantee the representativeness of the samples, we uniformly selected individual trees in space and number according to different landscape types. Specifically, we first delineated the boundaries of individual trees in the aerial photographs. Then, we further delineated individual trees' boundaries in CHMs according to the boundaries sketched in aerial photographs. The number of individual trees sketched in each landscape and the area of each landscape are shown in Table 1.

There are some small differences in the number of delineated individual trees between aerial photographs and LiDAR-derived CHMs, with more tree crowns sketched in CHMs than in aerial photographs. We chose trees that could be delineated in both the aerial photograph and CHM to validate the proposed approach, and meanwhile assigned a same ID number to the same trees by editing the vector in QGIS for both datasets, establishing which tree pairs matched correctly. In other words, a pair of trees are considered correctly matched when they have the same ID number. The percentage of correctly matched trees is used to evaluate the matching accuracy of individual trees for our proposed approach.

**Table 1.** The number of individual trees manually delineated from aerial photographs and LiDAR-derived CHMs, as well as its corresponding number of sample plots and total area in each landscape.

| Landscape | Number of Trees in Aerial Photographs | Number of Trees in CHMs | Number of Sample Plots | Total Area of Sample Plots (ha) |
|---|---|---|---|---|
| Broadleaved Forest | 788 | 829 | 8 | 11.93 |
| Coniferous Forest | 739 | 831 | 8 | 7.99 |
| Mixed Forest | 676 | 725 | 8 | 8.86 |
| Roadside Trees | 792 | 808 | 18 | 17.31 |
| Garden Trees | 680 | 776 | 7 | 16.07 |
| Parkland Trees | 768 | 783 | 7 | 18.84 |

*3.6. Visualization of Individual Tree Mismatches between Aerial Photographs and CHMs*

As shown in Figure 3, there are some examples of the spatial mismatch of individual trees in the manually delineated datasets, which are caused by local displacement between aerial photograph and CHMs. These mismatches are spatially inconsistent in both distance and orientation, and consequently cannot be directly addressed through traditional global geographic calibration for images [69,128–130].

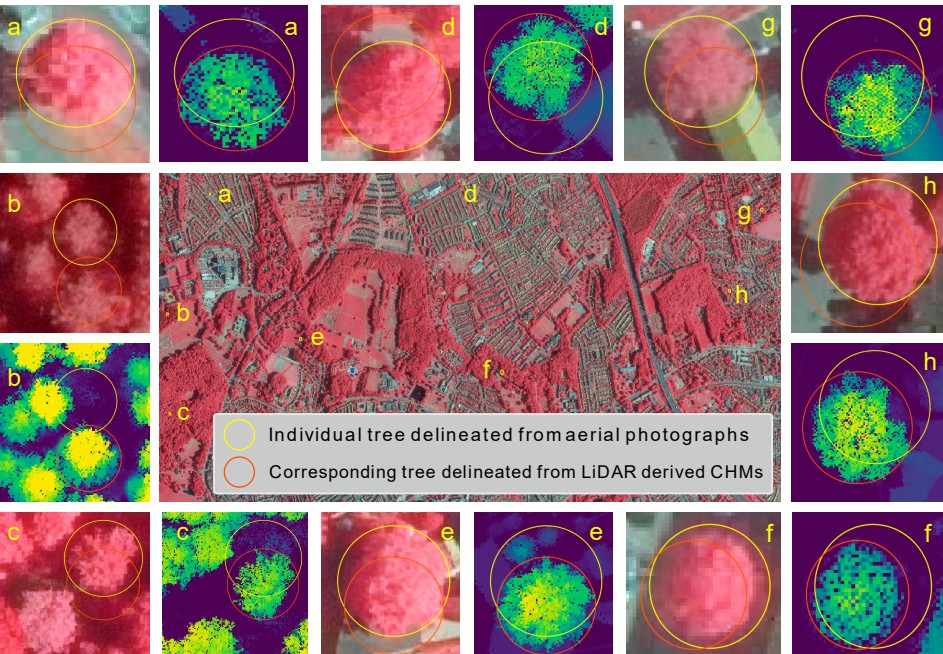

**Figure 3.** Examples of the spatial mismatches between individual trees resulting from local displacement between aerial photographs and the canopy height models (CHMs) generated from the airborne LiDAR point cloud data in the study area. Yellow and red circles with the same letter have the same geographical location.

*3.7. Statistical Description on Mismatch of Individual Trees*

We used crown area ratio and offset to explore the mismatch of individual trees between aerial photographs and CHMs (airborne LiDAR point cloud data were considered as the reference data in this study) (Figure 4). We defined the crown area ratio as the ratio of the crown area of an individual tree in the aerial photographs to the crown area of the corresponding tree in CHMs. The offset is defined as the distance between centers of two matching trees in aerial photographs and CHMs. We also performed a statistical description on the original offset of individual trees in each landscape (Figure S2).

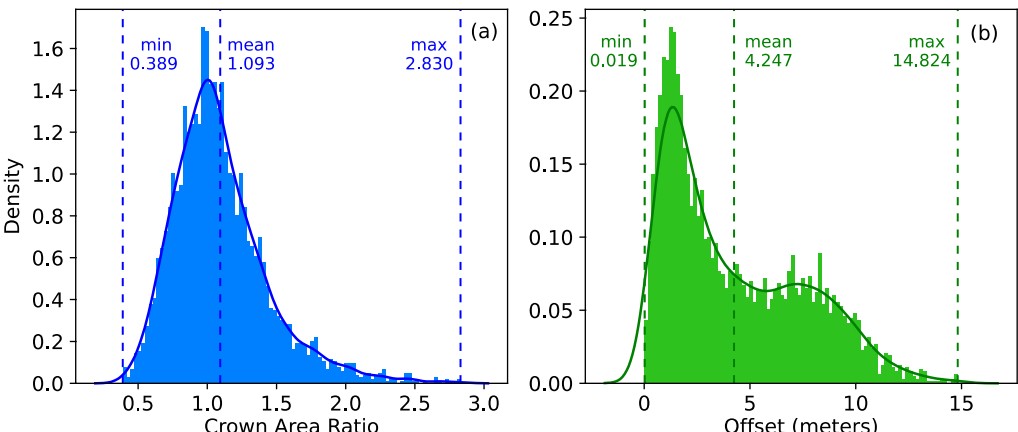

**Figure 4.** The statistical description on the mismatch of individual trees between aerial photographs and CHMs before rectifying the offset. (**a**) Density histograms of the crown area ratio of individual trees in aerial photographs to the corresponding trees in CHMs, and (**b**) density histograms of the offset between the center of two matching trees separately from aerial photographs and CHMs. The bar, vertical dotted line, and curve represent the density, mean, and density distribution curve of corresponding metrics, respectively.

### 3.8. Normalized Intersection over Union

The intersection over union (IoU, Figure 5a) is commonly used to evaluate the performance of an object detector for a specific dataset [131–133]. In the case of object detection and segmentation, IoU quantifies the degree of overlap between the target bounding box and the predicted bounding box [134–136]. The larger the overlapping area, the greater the IoU. The bounding boxes are usually the same size in most applications about object matching. While in our study, we found that the bounding box size between the tree in the aerial photograph and the corresponding tree in the CHMs was not always the same in most pairs of individual trees (Figure 4a), leading to discrepancies when evaluating the matching accuracy with IoU. In response, we proposed a normalized intersection over union (NIoU) to eliminate the impact of area difference on IoU by multiplying the ratio of the area of the larger tree to that of the smaller one (Figure 5b). The value of NIoU is between 0 and 1, the closer the value is to 1, the higher the degree of overlap between the two trees. We computed the IoU and NIoU for a roadside tree sample individually to show the difference in measuring the overlap between trees in both datasets (Figure 6). In this study, NIoU served two functions, i.e., (1) screening out the correct or optimal offset vector in determining both the correct offset of a particular tree and the final offset of trees in a region; (2) evaluating the matching accuracy after rectifying the offset of individual trees in two images.

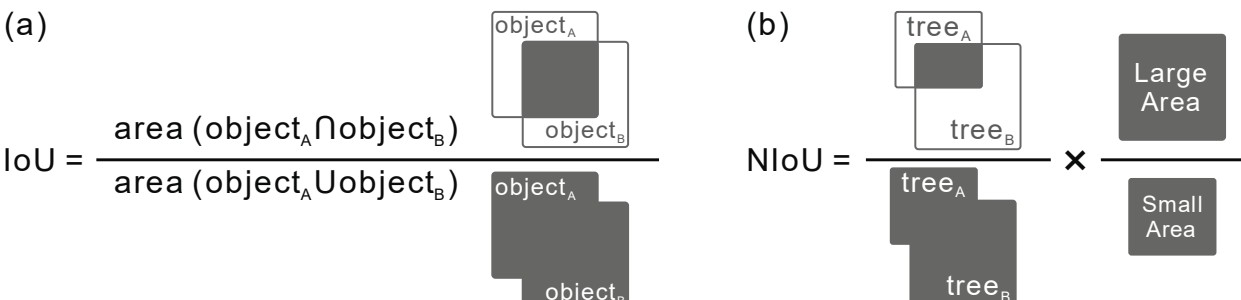

**Figure 5.** The formula and schematic diagram of intersection over union (IoU (**a**)) and normalized intersection over union (NIoU (**b**)).

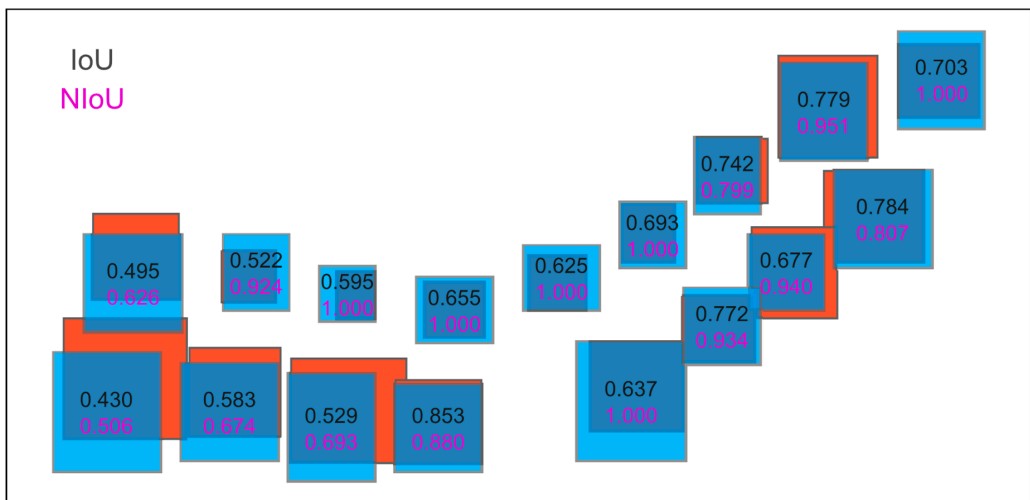

**Figure 6.** Comparison of IoU and NIoU in measuring overlap between trees. The numbers over the bounding boxes represent the corresponding values.

### 3.9. Accuracy Assessment

In our preliminary experiment, we found that some trees were assigned to non-matching trees when calculating the NIoU, especially in coniferous forest (Figure 3b). This is due to the fact that the algorithm considers the tree with maximum NIoU as the matching tree when calculating neighbor candidate trees with the reference tree. Therefore, we first used the pairing rate (the percentage of correctly paired trees) to screen out those trees that de facto were not pairwise matching with the reference tree (Equation (1)).

$$\text{pairing rate} = \frac{N_{same\_ID}}{N} \times 100\% \qquad (1)$$

where $N_{same\_ID}$ represents the number of tree pairs with the same ID number when calculating the NIoU of individual trees between two data, while $N$ is the total number of tree pairs. And then, we define matching accuracy (Equation (2)) as the mean NIoU calculated from the remaining matching tree pairs to measure the degree of overlap between the correctly paired trees.

$$\text{matching accuracy} = \frac{\sum NIoU}{N_{same\_ID}} \qquad (2)$$

where $\sum NIoU$ indicates the sum of NIoU calculated from each matching tree pair. Moreover, the density histograms with curves of NIoU calculated from matching tree pairs were exploited to compare the difference more intuitively between the results before and after rectifying the offset.

### 3.10. Parameter Tuning

The key parameter for our approach is the number of reference trees used for calculating, selecting, and determining the final offset vector. Hence, we tested the pairing rate and matching accuracy as the number of reference trees increases. We tested the number of reference trees from one to ten. Depending on the number of reference trees we needed to test, we randomly selected the specific number of trees from CHMs as reference tree to calculate their correct offset vectors, and then chose the final one to rectify the trees in aerial photographs (the core steps are the same as our proposed approach in Section 2.1). We repeated the above experimental steps with 20 different random seeds in all samples of six landscapes (Figure 1b) and calculated the average value of the pairing rate, matching accuracy, and their standard deviation.

## 4. Results

### 4.1. Parameter Tuning

As shown in Figure 7a, when the number of reference trees reaches three, the pairing rate is close to saturation and its standard deviation (SD) no longer significantly changes. While the matching accuracy and its SD becomes steady after the number greater than eight (Figure 7b). Therefore, in accordance with the "maximum principle", we set the number of reference trees to be eight to apply our proposed method if there are too many reference trees for selecting the correct offset vectors.

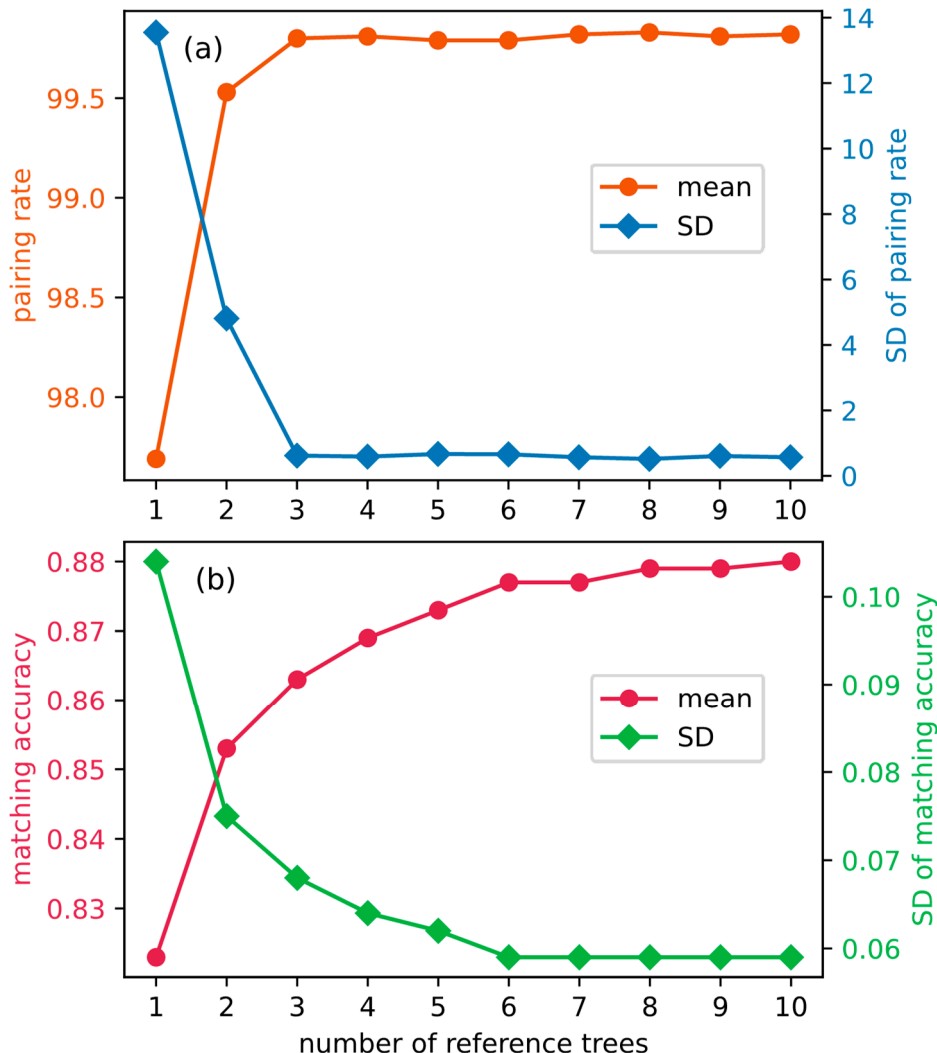

**Figure 7.** The evaluation indices and their standard deviation (SD) of individual trees using our proposed matching approach as the number of reference trees increases. (**a**) pairing rate; (**b**) matching accuracy.

### 4.2. Pairing Rate after Using Our Proposed Approach

The pairing rate of the individual trees in all six landscapes before and after rectifying the offset is presented in Figure 8, which shows that all trees were correctly paired after offset rectification using our proposed approach. The mean pairing rate of all six landscapes increased from 71.13% (SD = ±29.83) to 100.00% (SD = 0). Specifically, coniferous forest had the largest improvement with an increase of 76.54%, followed by broadleaved forest (55.03%), mixed forest (37.96%), and garden trees (3.70%), while the individual tree pairs in roadside trees and parkland trees were totally paired even before offset rectification.

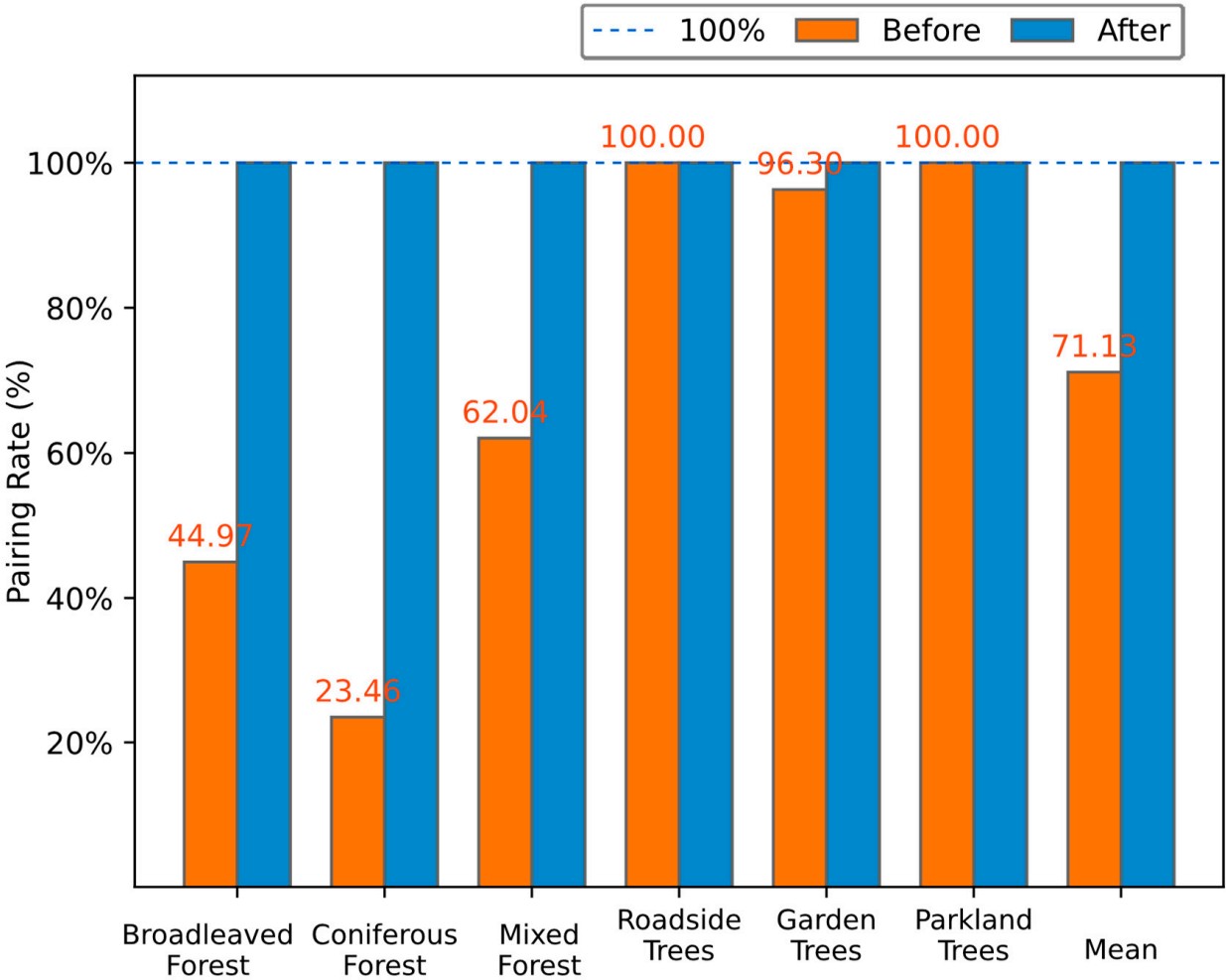

**Figure 8.** The pairing rate of the individual trees in all six landscapes before and after offset rectification.

### 4.3. Matching Accuracy after Using Our Proposed Approach

Figure 9 details the density of NIoU calculated from matching tree pairs in all six landscapes before and after rectifying the offset. In all six landscapes, the matching accuracy (mean NIoU) improved significantly ($p < 0.001$, ANOVA), and coniferous forest demonstrated the greatest improvement with an increase of 0.546, followed by mixed forest (0.469), broadleaved forest (0.391), roadside trees (0.218), garden trees (0.178), and parkland trees (0.091). The optimal matching accuracy of individual trees before and after offset rectification was respectively in parkland trees (0.796) and roadside trees (0.919), which was also the only landscape with matching accuracy greater than 0.9. Before applying our approach, the mean matching accuracy of landscapes in city (Figure 9d–f) was 0.718, which was significantly higher than that in forest (0.362). While after rectifying the offset, the six landscapes with matching accuracy in descending order were sequentially roadside trees (0.919), parkland trees (0.887), broadleaved forest (0.853), garden trees (0.835), mixed forest (0.823), and coniferous forest (0.815). Overall, the mean matching accuracy of all six landscapes significantly rose from 0.642 ± 0.264 (SD) to 0.861 ± 0.152 (SD).

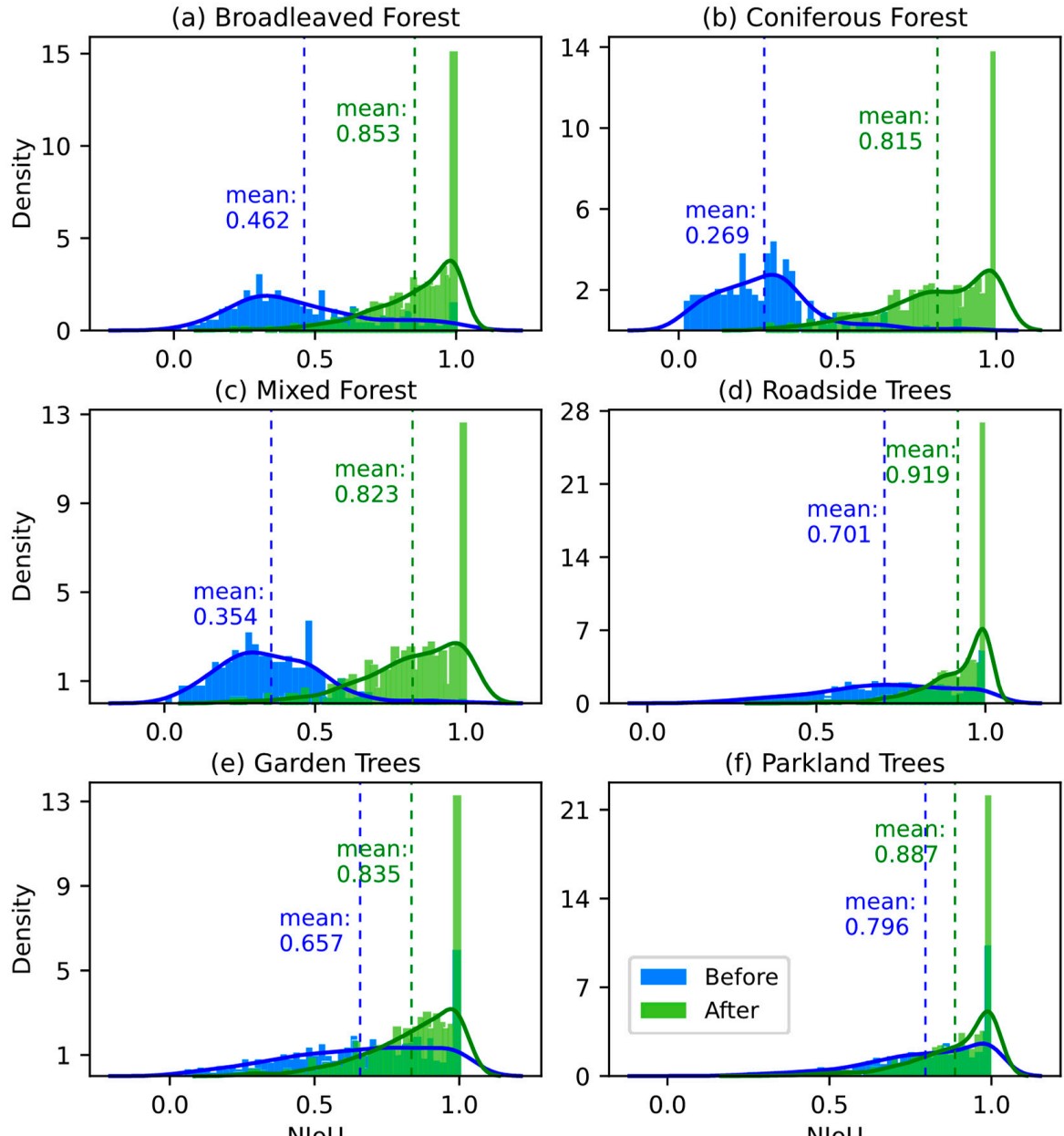

**Figure 9.** The overlaid density histograms with density curves of NIoU before and after offset rectification. The bar and curve represent the density and its distribution curve of NIoU, respectively, while the vertical dash line is the mean NIoU (i.e., matching accuracy). The blue elements denote the results before offset rectification, while the results after rectifying the offset are shown in green.

*4.4. Visualization of Pairing Rate and Matching Accuracy before and after Rectification*

We selected a representative sample from each landscape to visually illustrate the pairing and matching effect of individual trees before and after offset rectification (Figure 10). As observed in Figure 10, the pairing rate of the individual trees in all six landscapes was improved to 100% following application of the new matching algorithm described above, while the pairing rate was 14.65%, 52.60%, and 88.59% in broadleaved, coniferous, and mixed forest, respectively, before rectifying the offset (Figure 10a–c). Meanwhile, mean matching accuracy increased to 0.839. Regarding the roadside, garden, and parkland trees, although the pairing rate has not increased significantly, the matching accuracy of roadside and garden trees has been substantially raised (Figure 10d,e). Furthermore, the matching accuracy in parkland trees was slightly improved after offset rectification (Figure 10f).

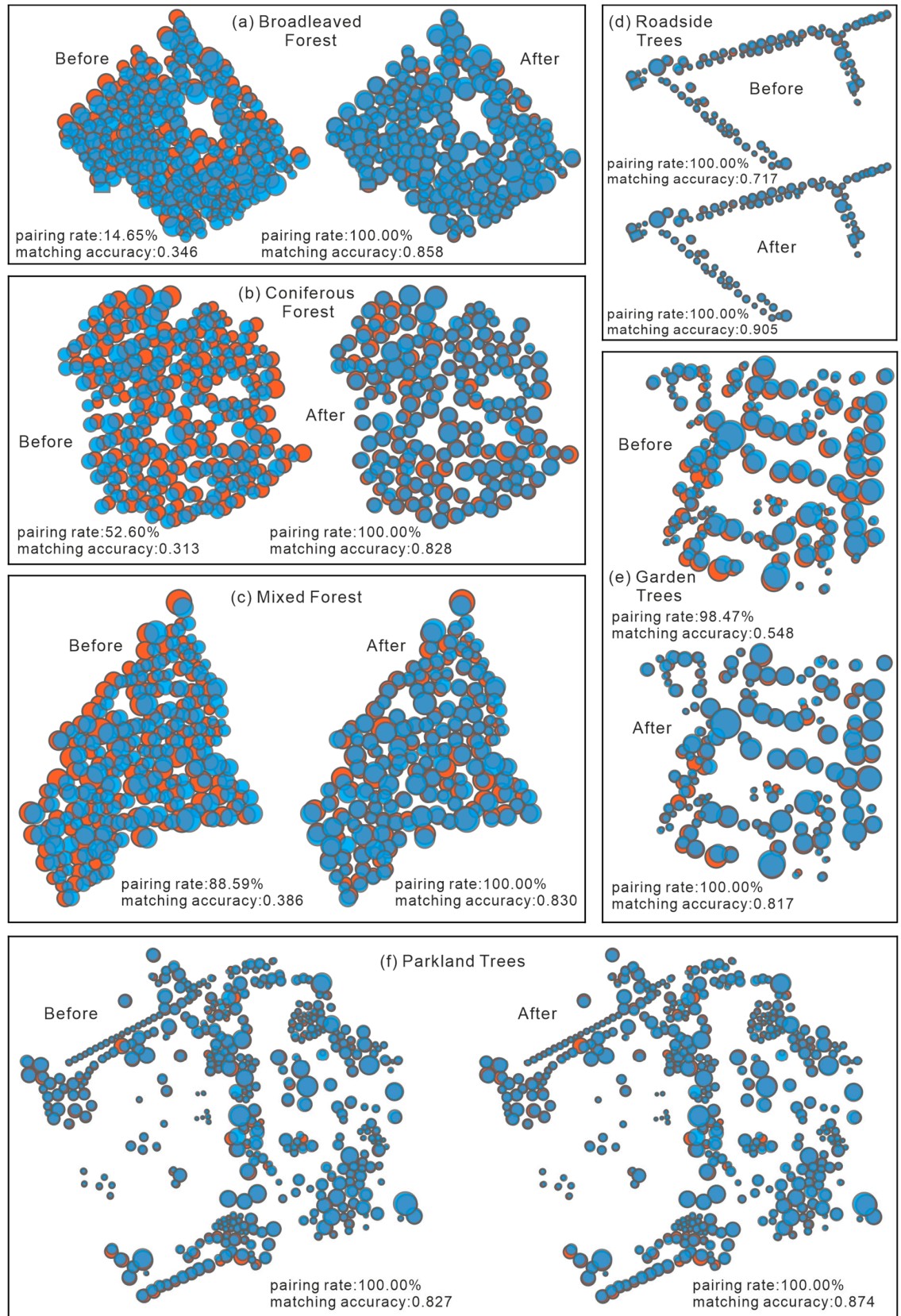

**Figure 10.** Instance of visualization of the pairing and matching effects of individual trees in all six landscapes before and after offset rectification.

*4.5. Comparison of Results Using Image Registration with Our Proposed Approach*

We compared the pairing rate and matching accuracy of individual trees after registering images with ENVI automatic registration module and matching with our proposed approach, respectively. As shown in Table 2, through our proposed approach, the mean pairing rate is up to 100.00%, while about 10% of the trees remain mismatched when using the conventional image registration method. In terms of matching accuracy, our proposed approach achieved better results and had smaller standard deviations. It is noted that the matching accuracy in urban areas (roadside trees, garden trees, and parkland trees) has a certain degree of decline, although the pairing rate increases. Moreover, comparing the evaluation indices of the traditional method and our proposed approach, the *p*-values of the paired *t*-test for pairing rate and matching accuracy were 0.045 and 0.017, respectively.

**Table 2.** The pairing rate, matching accuracy, and its standard deviation (SD) of each landscape using image registration module (ENVI) and our proposed approach.

| Landscape | Image Registration (ENVI) | | Our Proposed Approach | |
|---|---|---|---|---|
| | Pairing Rate | Matching Accuracy (SD) | Pairing Rate | Matching Accuracy (SD) |
| Broadleaved Forest | 88.98% | 0.710 (0.196) | 100.00% | 0.853 (0.175) |
| Coniferous Forest | 99.75% | 0.788 (0.186) | 100.00% | 0.815 (0.178) |
| Mixed Forest | 100.0% | 0.797 (0.181) | 100.00% | 0.823 (0.171) |
| Roadside Trees | 79.13% | 0.656 (0.153) | 100.00% | 0.919 (0.100) |
| Garden Trees | 92.94% | 0.682 (0.206) | 100.00% | 0.835 (0.199) |
| Parkland Trees | 87.06% | 0.676 (0.157) | 100.00% | 0.887 (0.147) |
| Mean | 91.31% | 0.692 (0.175) | 100.00% | 0.861 (0.152) |

## 5. Discussion

*5.1. Influence of Registration Noise on Pairing Rate*

Results of this study show that the pairing rate of individual trees in the deciduous, coniferous, and mixed forest classes was clearly lower than that for roadside, garden, and parkland trees (Figure 8). We can readily infer that registration noise exerts a greater influence on forest classes, which might be caused by the quantity and quality of ground control points available for geo-correcting aerial photographs. Compared to roadside, garden, and parkland trees, it is more challenging to locate highly precise control points for fine registration in forested areas (Figure 2), and this leads to a larger offset between individual trees of two data. In general, the lower the density of trees, the higher the matching accuracy and vice versa [137]. It is not a unique problem to this study as there is generally a lack of tie points in areas that are difficult to survey (e.g., mountains, grasslands, and forests) [138]. There are only a few trees in the garden class that were not calculated from matching tree pairs, probably due to the larger variance in canopy and tree height from a higher abundance of tree species (Figure 2c(V)).

*5.2. Effectiveness of the Proposed Approach*

The improvement of pairing rate was more prominent in coniferous forest, probably since coniferous trees typically had large spaces between adjacent trees than broadleaved trees [25,139,140]. As shown in Figure 3b, coniferous forests were prone to be reference trees to compute the NIoU with a non-matching tree once the offset exceeds the diameter of the tree (especially in forest with high-density trees). In comparison, broadleaved forest and mixed forest had a relatively better pairing rate. Moreover, the improvement of matching accuracy in forest was more evident than that in landscapes of city. The matching accuracy of individual trees in forest before rectifying the offset was much lower than for roadside, garden, and parkland trees. After offset rectification, the matching accuracy reached an asymptotic and highest level for all six landscapes, demonstrating the effectiveness of our

proposed matching approach in improving matching accuracy regardless of landscape type (see Figure 9).

### 5.3. Matching Accuracy in Different Landscapes

As described above for the pairing rate, the most dramatic improvement in the matching accuracy was for coniferous forest. Except for the greater offsets in forests, smaller crowns of coniferous trees had smaller NIoU values, as expected from the NIoU formula (Figure 5). Therefore, in forests, the matching accuracy of broadleaved forest ranks first, followed by mixed forest, and finally coniferous forest. For roadside, garden, and parkland trees, the maximum increment and the maximum value of matching accuracy after rectifying the offsets were for the roadside trees. Several possible reasons may have contributed. Most roadside trees consist of broadleaved trees which were usually the same species, similar in tree age (i.e., tree height and crown size) [141–143]. The regular layout of roadside trees made the offset simple and spatially consistent [144,145], reducing the local registration noise for roadside trees.

The best matching accuracy was observed in different landscapes before and after offset rectification. Before rectifying the offsets between individual trees from two datasets, the best matching accuracy was observed in parkland trees, one possible reason is that the design and context of parklands provide richer features for geographic registration of aerial photographs, as well as parklands had large widely spaced trees separated by grass areas [146,147]. Moreover, although garden trees were characterized by highly accurate ground control points due to their (mostly) solitary and isolated nature and growing environments surrounded by roads and buildings, tree matching results were not as satisfactory as expected, probably stemming from the species-rich areas with trees varying in height and crown size [148–151], complicating the degree and orientation of offsets between different trees.

### 5.4. Analyzing the Results Using Conventional Image Registration Approach

Compared with that before image registration, the matching accuracy has been improved for forest landscape (broadleaved forest, coniferous forest, and mixed forest), but there has been a certain degree of decline in urban landscapes (roadside trees, garden trees, and parkland trees) (Table 2). Meanwhile, the pairing rate of the corresponding landscape had a similar trend. The above phenomenon was mainly caused by unevenly distributed, low precision, and insufficient control points (CPs) that were found by ENVI automatic program. In this study, we chose landscapes in small patches to verify our proposed method, while it was difficult to find suitable and sufficient CPs to align images with high accuracy [152–155]. Taking the landscape roadside trees as an instance, its area is always narrow and long, thus the unevenly distributed points can easily change the shape of the image to be registered, resulting in serious distortion of the edge of the image. Therefore, though many methods have been proposed to solve multimodal images registration problems, they are usually suitable for large-scale images due to the uneven distribution and limited quantity of CPs [156,157], which is opposed to solving local registration noise.

In our study, aerial photographs and CHMs (different modalities) make it more difficult to find high-precision CPs. On the one hand, the values in different images represent different meanings, e.g., individual trees with high optical reflectance do not always correspond to high altitude in LiDAR height products and vice versa. On the other hand, different viewpoints also result in different shapes for the same objects, e.g., area (aerial photographs) to line (CHMs), line (aerial photographs) to point (CHMs). In addition, the characteristics of ground features change with time. Moreover, pixel values in CHMs are not always geographically true values and some values are generated by interpolation of adjacent values due to missing or insufficient points of the corresponding location in LiDAR point cloud data [158–160]. Therefore, looking for evenly distributed and high-quality CPs between multimodal images requires a large amount of manual intervention, while the method is usually not applicable for large-scale area and fine-grained object

studies. In conclusion, in the face of different area sizes, our proposed matching approach is more robust.

### 5.5. Choosing a Suitable Threshold for Specific Applications

In the object detection domain, a predicted bounding box is widely considered to be correctly detected if its IoU with the ground truth bounding box is greater than 0.5 [1,133,134]. In this study, we used NIoU to determine whether the candidate tree was correctly matched with the reference one, and we found the widely accepted threshold of NIoU (i.e., 0.5) was not always optimal. Meanwhile, other factors like stand density and crown size of individual trees ought to be considered. For instance, in roadside trees where there was sufficient space between adjacent trees, we found an NIoU of around 0.4 or less. In a nutshell, the choice of an appropriate NIoU threshold for a specific application was required, which [161] also recommended. Last but not the least, in theory, the smaller the area of individual trees to be matched, the higher the matching accuracy. However, the corresponding amount of computation will also increase substantially, which should not be overlooked for large areas.

### 5.6. Possible Challenges and Improvements in Practical Application

First, one apparent challenge comes from the quality of individual tree products. Unlike the manual delineated trees used in this study, the over- and under- segmentation are unavoidable phenomena when using algorithm to automatically extract boundary information of individual trees, which can easily lead to some trees not having the corresponding trees in another dataset. While the rapid advancement of deep learning promises opportunities to reduce the errors caused by the phenomena. Another possible challenge may arise in plantation forests, the offset vector might be wrongly chosen when the spatial layout of trees is similar. At this point, we can expand the range of candidate trees to alleviate achieving the wrong offset vector. Moreover, we can make an improvement of our proposed approach by sightly modifying. If we need to achieve a more accurate or even completely corresponding offset vector for each tree, we can narrow down the trees that used to confirm which offset vector is optimal. However, it would cause a large amount of computation and is not suitable for large-scale use, like national or continental scales.

### 6. Conclusions

In this study, we proposed a novel tree-oriented matching algorithm that improved the pairing rate and matching accuracy of individual trees derived from aerial photographs and airborne LiDAR point cloud data. Our results demonstrated that the proposed approach effectively increases the pairing rate and matching accuracy of individual trees that were manually delineated from aerial photographs and LiDAR-derived CHMs. Compared to the traditional registration method, the average pairing rate of individual trees for all six landscapes increased from 91.13% to 100.00% ($p = 0.045$, $t$-test), which suggested that our proposed approach could perfectly solve the problem of matching trees. Meanwhile, the average matching accuracy increased from $0.692 \pm 0.175$ (standard deviation) to $0.861 \pm 0.152$ ($p = 0.017$, $t$-test), demonstrating the effectiveness of the proposed matching approach in matching individual trees between multimodal images.

**Supplementary Materials:** The following supporting information can be downloaded at: https://www.mdpi.com/article/10.3390/rs15174128/s1. Figure S1: Aerial photographs respectively acquired in 2018 (a) and 2020 (b). Figure S2: The original offsets of individual trees between aerial photographs and CHMs derived from airborne LiDAR data in six landscapes.

**Author Contributions:** Conceptualization, methodology, programming, validation, visualization, writing—original draft preparation, Y.X.; formal analysis, investigation, Y.X. and T.W.; supervision, T.W. and A.K.S.; writing—review and editing, T.W., A.K.S. and T.W.G.; funding acquisition, Y.X. and A.K.S. All authors have read and agreed to the published version of the manuscript.

**Funding:** The China Scholarship Council (202008440522) and the ITC Research Fund co-funded this research. The work was also supported by the European Union's Horizon 2020 research and innovation program (834709), funded by the European Research Council (ERC).

**Conflicts of Interest:** The authors declare no conflict of interest.

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
