# Peer review of "A Novel Approach to Match Individual Trees between Aerial Photographs and Airborne LiDAR Data"

_remotesensing, doi:10.3390/rs15174128_

Round 1
Reviewer 1 Report
The topic is well introduced. Your effort to propose a procedure that enables a better alignment of trees identified in different RS products is appreciable.
Nonetheless I actually could not follow the algorithm you expose.
Moreover, using manual procedure to delineate with circles the individual crowns, particularly having ‘informally’ “uniformly selected individual trees in space and number”, doesn’t appear as a reliable ‘random sample’.
The evaluation of the result, if I understand correctly, is based on the rectification of the elements used to compute the rectification parameters. If so the figures exposes would evidently not be useful.
In the attached pdf you can find more detailed comments, including some notes concerning the minor English language problems.

see above
Reviewer 2 Report
This research proposes a multi-modal data matching method (LiDAR-Aerial image) for single tree objects, which is innovative to a certain extent, and has been applied to 6 landscape types of sample plots, and the matching accuracy has been improved, laying a foundation for the subsequent multi-modal data fusion, and providing scientific and technical support for the efficient management of forestry science. It’s a interesting work over all.However, there are still the following points to be further modified and improved:
1. Lines 53-79:In the introduction part, the author uses a lot of space to introduce the advantages of multi-modal fusion data, but in fact, the research in this paper is to solve the problem of location mismatch of multi-source data before fusion, and it does not achieve fusion. It is suggested to add more summary and elaboration of existing research on matching.
2. Lines 197-198:The author tested and verified the proposed method by using artificially marked tree crown width as sample validation. There were human errors in this marking method, but the accuracy of the marking results was not verified. In fact, the proposed method depended on the marked crown width area.
3. Lines 207:The ranges of Area (ha) and Range of area (ha) in Table 2 are strange. Please distinguish and explain them for better understanding.
4. Lines 355-365: the author only evaluated the accuracy of the results of the proposed method, and found that the evaluation of the time efficiency of this method, and only selected the matching results of the automatic image matching module in ENVI for comparison, whether it is insufficient to reflect the superiority of the proposed method, it is suggested to increase the comparison with other methods.
5. There are formatting errors in the references, please check all references, such as the year should be bolded and the journal name italicized.
This research proposes a multi-modal data matching method (LiDAR-Aerial image) for single tree objects, which is innovative to a certain extent, and has been applied to 6 landscape types of sample plots, and the matching accuracy has been improved, laying a foundation for the subsequent multi-modal data fusion, and providing scientific and technical support for the efficient management of forestry science. It’s a interesting work over all.However, there are still the following points to be further modified and improved:
1. Lines 53-79:In the introduction part, the author uses a lot of space to introduce the advantages of multi-modal fusion data, but in fact, the research in this paper is to solve the problem of location mismatch of multi-source data before fusion, and it does not achieve fusion. It is suggested to add more summary and elaboration of existing research on matching.
2. Lines 197-198:The author tested and verified the proposed method by using artificially marked tree crown width as sample validation. There were human errors in this marking method, but the accuracy of the marking results was not verified. In fact, the proposed method depended on the marked crown width area.
3. Lines 207:The ranges of Area (ha) and Range of area (ha) in Table 2 are strange. Please distinguish and explain them for better understanding.
4. Lines 355-365: the author only evaluated the accuracy of the results of the proposed method, and found that the evaluation of the time efficiency of this method, and only selected the matching results of the automatic image matching module in ENVI for comparison, whether it is insufficient to reflect the superiority of the proposed method, it is suggested to increase the comparison with other methods.
5. There are formatting errors in the references, please check all references, such as the year should be bolded and the journal name italicized.
Round 2
Reviewer 1 Report
Excuse me, as stated in advance, I can not dedicate much time to the revision.
If it can be of any help I try to report o my quick evaluation.
With your revision I understand al little bit better the algorithm but I still have questions
Example sentence "In the face of the challenges posed by typical remote sensing image registration methods, researchers have turned to the transformative potential of deep learning techniques in the realm of computer vision." that is understandable but I think should be rephrased.
Do you mean: "Researchers dealing with the recognition and delineation of objects in images are considering to exploit deep learning techniques developed for computer vision objectives"?
